# An Investigation of the Influence of Age and Saliva Flow on the Oral Retention of Whey Protein and Its Potential Effect on the Perception and Acceptance of Whey Protein Beverages

**DOI:** 10.3390/nu12092506

**Published:** 2020-08-19

**Authors:** Victoria Norton, Stella Lignou, Stephanie P. Bull, Margot A. Gosney, Lisa Methven

**Affiliations:** 1Department of Food and Nutritional Sciences, University of Reading, Whiteknights, Reading, Berkshire RG6 6AD, UK; v.l.norton@pgr.reading.ac.uk (V.N.); s.lignou@reading.ac.uk (S.L.); s.p.bull@reading.ac.uk (S.P.B.); 2Royal Berkshire NHS Foundation Trust, London Road, Reading RG1 5AN, UK; m.a.gosney@reading.ac.uk

**Keywords:** mucoadhesion, mouthdrying, older adults, whey protein, saliva flow

## Abstract

Protein fortified products are regularly recommended to older adults to improve nutritional status and limit sarcopenia. However protein fortification can elicit negative sensory attributes such as mouthdrying. Sensitivity to mouthdrying can increase with age, yet the influence of saliva flow and mucoadhesion remain uncertain. Here, two studies tested different whey protein beverages (WPB); 22 healthy younger volunteers completed a pilot and 84 healthy volunteers from two age groups (18–30; 65+) completed the main study. In both studies salivary flow rates (mL/min) were measured and saliva samples were collected at time intervals post beverage consumption to measure mucoadhesion to the oral cavity, where protein concentration was analysed by Bradford Assay. Volunteers rated perception and acceptability of WPBs in the main study. WPB consumption resulted in significantly increased protein concentration (*p* < 0.0001) in saliva samples compared with a control whey permeate beverage. Older adults had significantly lower unstimulated saliva flow (*p* = 0.003) and significantly increased protein concentration (*p* = 0.02) in saliva samples, compared with younger adults. Heating of WPB significantly (*p* < 0.05) increased mouthdrying and thickness perception and reduced sweetness compared with unheated WPB. Mucoadhesion is concluded to be a true phenomenon in WPBs and increases with age.

## 1. Introduction

Malnutrition is prevalent within the UK, with over one million older adults affected, and the risk of malnutrition is considered to increase in clinical settings (hospitals, care homes and mental health units) [1]. Such malnutrition has multiple contributing factors and can refer to an overall insufficient intake of all nutrients or specific macro- or micronutrients [1]. Protein is of specific interest as protein needs are considered to increase with age, for example, the PROT-AGE study group recommend a protein intake of 1.0–1.2 g/kg/d for older adults [2]. They suggested that this higher requirement was to maintain good health, encourage recovery from illness and preserve functionality; and that increased needs of older adults compared with younger adults resulted from age-related changes in protein metabolism [2]. The intake of food generally, and of protein rich foods specifically, can be reduced in older age due to chemosensory impairments, such as loss of taste and smell, which are commonly associated with older adults and considered to relate to ageing, medication, disease, malnutrition, environment and surgical interventions [3]. However, the influence of saliva, and age-related changes in saliva, on the sensory perception of foods and protein fortified foods, has received little attention.

Saliva is a viscoelastic solution, consisting of 99% water, with the remainder being protein and ion components, which enable taste, aid digestion and prevent tooth decay, as well as acting as a lubricant and having antimicrobial properties [4]. Saliva flow is considered to decrease with age [5], accordingly, a reduced saliva flow is considered a problem and is commonly associated with decreased lubrication, protection, oral clearance, mucosal surface hydration and coating abilities within the oral cavity [6,7,8,9]. Furthermore, food breakdown and perception of taste, flavour and texture of foods are all influenced by saliva, impacting upon the eating process and food intake [10,11]. This emphasises the need to understand how saliva can impact sensory perception and consumption of foods in older adults.

Foods for older adults are often fortified by whey protein due to the high bioavailability of this protein source [12]. To increase energy, protein and micronutrient intake, oral nutritional supplements (ONS) are often prescribed to older adults, and these are commonly fortified with whey protein, and other dairy proteins sources. However, ONS typically have poor consumer acceptance, which has been linked to both undesirable taste and aroma and a build-up of perceived mouthdrying following repeated consumption [13,14,15,16]. Previous research has shown that older adults have greater sensitivity to mouthdrying compared with younger adults following consumption of dairy beverages [17]. Sensory profiling has identified negative mouthfeel attributes to be perceived after consumption of whey protein (e.g., chalky, drying, mouthcoating, astringent) and heat treatment of whey protein is considered to intensify further these sensory properties [18]. Such findings have clinical significance, particularly as individuals are commonly recommended to consume up to 600 mL of ONS per day [16]. Astringency, drying and mouthdrying are terms commonly used to describe the ‘textural defects’ associated with dairy products [19]; these terms are often used within the literature as interchangeable. Astringency typically refers to a mouth puckering like sensation caused by precipitation of salivary proteins on binding to polyphenols which reduces salivary lubrication [20]; however polyphenols (a group of secondary plant metabolites) [21] are not present in whey protein. In this paper the term mouthdrying is used to refer to a drying sensation in the mouth during or after the consumption of a product. The causes of whey protein derived mouthdrying are currently not fully understood, despite previous investigation and are summarised in Table 1.

Our research group consider the adhesion of whey protein to be a highly probable cause of whey protein derived mouthdrying. Mucoadhesion, a concept that has been utilised in drug delivery systems [35,36,37,38] and more recently considered in a food context [39]. Mucoadhesion can be described as physicochemical interactions between a polymetric material and mucosal environment [38] and is considered in the context of this paper to be the binding or sticking of whey proteins to the oral cavity [18]. An oral retention method has been developed to measure the amount of protein retained in the mouth over time by measuring protein in saliva samples with factors such as salivary flow, composition and viscosity considered to influence retention of samples [33,40]. One limitation of the oral retention method of measuring protein mucoadhesion to date has been small subject sample sizes [33]. Currently, the extent to which mucoadhesion and mouthdrying are influenced by saliva flow in older adults remains uncertain. This paper hypothesises whey protein beverages (WPB) will cause mucoadhesion of protein to the oral cavity following consumption, and that older adults will have reduced salivary flow, greater adhesion of protein to the oral cavity, and increased mouthdrying perception of WPB, when compared with younger adults. This hypothesis was tested through the following objectives:(1)A pilot study was carried out with the objective of establishing whether the protein measured in the oral cavity post WPB consumption resulted from mucoadhesion of the WPB (rather than resulting from consumption-induced release of salivary protein). The pilot study was conducted in younger adults and measured protein concentration of saliva samples post beverage consumption (WPB and whey permeate beverage (WPeB)) at 4 different timepoints (15 s, 30 s, 60 s and 120 s), in order to validate the oral retention method.(2)The main study had the following objectives: (a) to measure salivary flow rates from unstimulated and stimulated saliva, (b) to determine if protein adheres to the oral cavity of older adults to a greater extent than younger adults, (c) to determine if salivary flow rates influence mucoadhesion and perception of WPBs, and (d) to evaluate whether heat treatment of protein in WPB causes mouthdrying and reduced acceptance within each volunteer group. This study recruited younger and older adults to test these objectives.

## 2. Materials and Methods

### 2.1. Overview of Pilot and Main Study

The pilot study was a single blinded randomised crossover trial with one study visit, involving 22 healthy male and female younger volunteers (18–30 years; 25.7 ± 3.0 years). The main study consisted of 84 healthy male and female volunteers from two age groups (42 younger adults; 18–30 years, 24.3 ± 3.6 years and 42 older adults; over 65 years, 73.6 ± 6.2 years) who completed a single blinded randomised crossover trial involving three study visits (volunteer overview; Appendix A). In both studies the subject size was determined by power calculations (alpha risk = 0.05 and 80% power) based on previous study data [33] using protein retention in the oral cavity as the primary outcome measure. In the pilot study comparing WPB with WPeB we estimated a difference in protein concentration of 1.5 mg/mL saliva and standard deviation of 1.5, which concluded a minimum sample size of 15. In the main study to compare oral retention for WPB in younger versus older adults we anticipated a smaller difference of 0.7 mg/mL (standard deviation 1.5), inferring a minimum sample size of 72. Volunteers were recruited from the local Reading area. The studies were conducted in accordance with the Declaration of Helsinki. All volunteers had the study fully explained to them and provided informed written consent before taking part. They were informed that all data would be anonymous and kept fully confidential and that there was a right to withdraw. The studies received a favourable opinion for conduct from the University of Reading Research Ethics Committee (pilot study: SCFP 28/19 and main study: UREC 18/46) and the study was registered on the clinical trials database (www.clinicaltrials.gov as NCT03798730).

All volunteers were screened to ensure suitability (minimal medication, non-smoker, no food allergies or intolerances, non-diabetic and not having had either cancer, oral surgery or a stroke). Volunteers who met the inclusion criteria and were willing to take part were invited to attend study visits held at the Sensory Science Centre, University of Reading; the study overview is summarised in Figure 1. In order to control extraneous variables, volunteers refrained on the day of each study visit from tea and coffee and drank a glass of water one hour before the visit. Each individual volunteer completed all their study visits at the same time of day in a temperature-controlled room (22 °C) under artificial daylight.

### 2.2. Materials

Whey powders were provided by Volac (Volac International Ltd., Royston, UK) as whey protein concentrate (WPC, Volactive Ultra-Whey 80, providing a minimum protein content of 80% and the remaining 20% being lactose, fat, moisture and ash), as well as whey permeate (WPe, Volactose Taw Whey Permeate, providing a minimum lactose content of 89% and the remaining 11% being ash, moisture, protein and fat). Parafilm^®^, Bradford reagent (500 mL, 0.1–1.4 mg/mL) and protein standard 2 mg/mL (Bovine Serum Albumin, BSA) were supplied by Sigma-Aldrich (Dorset, UK).

### 2.3. Liquid Model Preparation

The pilot study tested two different beverages, first, a WPB (10% *w/v*, WPC powder in deionised water) was used as a protein beverage. A 10% concentration is considered sufficient to stimulate a postprandial muscle protein synthesis response in older adults and has previously been used in WPB testing [18,41]. Second, a WPeB (4% *w/v*, whey permeate powder in deionised water) was used as the control beverage. The WPeB concentration was selected as being below the lactose sweet recognition threshold (4.19%), therefore unlikely to cause sweetness-stimulated additional saliva flow and having a relatively similar mineral profile to the protein beverage [42,43]. The main study tested two different WPBs (10% *w/v*, WPC powder in deionised water, as outlined above) for the influence of heat treatment; using unheated WPB (WPCU) and heated WPB (WPCH) samples, an overview of both studies beverages is outlined in Table 2. Sample heating temperature (70 °C) was chosen as beta-lactoglobulin, the most abundant protein in WPC, has a critical temperature for denaturation of 70 °C [18,44,45]. The sample heating time of 20-min was selected as the maximum time the product could be maintained at 70 °C without aggregation and remain acceptable to serve to consumers [18]. In both the pilot and the main study, the method was adapted from previous work [18] with both samples being prepared simultaneously, as summarised in Figure 2.

#### 2.3.1. Salivary Flow Rates

Saliva collection methods were adapted from previous studies [5,46]. In the pilot study saliva collection (unstimulated and stimulated saliva) was carried out at the beginning of the study visit with approximately 10 to 15-min break between collection methods. During the main study unstimulated saliva was collected by volunteers at the beginning of all three study visits and two replicates of stimulated saliva were collected from volunteers during study visit one, with approximately 10 to 15-min break between each collection. The rationale for the saliva collection was based on unstimulated saliva being considered a baseline measure and potentially more influenced by age than stimulated saliva [47,48]. In addition, it was unrealistic to expect older volunteers to provide a total of 10 saliva samples during a single study visit, therefore it was considered impossible to collect both unstimulated and stimulated saliva at study visits two and three. However, stimulated saliva was used as a baseline value for saliva samples post beverages consumption (see Section 2.3.2) as stimulated saliva is produced during food mastication and has supported better correlations with study outcomes compared with unstimulated saliva [48,49] and accordingly has been used in other saliva studies for analytical saliva analysis [50].

Unstimulated saliva was collected by asking volunteers to collect saliva in their mouths and to spit out saliva every time they felt the urge to swallow during a 5-min time period; saliva was collected in a wide lid collection tube (60 mL). Stimulated saliva was collected by asking volunteers to spit out saliva every time they felt the urge to swallow during a 2-min time period, while chewing on parafilm^®^ (5 × 5 cm), again into a wide lid collection tube (60 mL). Saliva weights for all volunteers were monitored by weighing collection tubes before and after collection. Using the weights collected, salivary flow rates were calculated as mL/min, using the assumption 1 g of saliva equates to 1 mL. All saliva samples were stored on ice pending analysis.

#### 2.3.2. Saliva Samples Post Beverage Consumption

An adapted oral retention method (Figure 3) [33,40] was used in both studies to measure the protein remaining in saliva samples post beverage consumption. Stimulated saliva samples were collected (as described above) and used as a baseline measurement. All volunteers were provided with eight 10 mL beverage samples (pilot study: 4 ×WPB and 4 × WPeB; main study: 4 × WPCU and 4 × WPCH) monadically in a balanced order; all samples were presented in opaque black plastic cups (25 mL) (BB Plastics, West Yorkshire, UK) (to mask minor visual differences between samples) coded with a random three-digit number. Volunteers also gave eight saliva samples post beverage consumption at defined randomised time points (15 s, 30 s, 60 s and 120 s). The procedure was carried out in duplicate for all volunteers (visits two and three) during the main study. In order to prevent crossover effects, volunteers had an enforced 5-min break between samples where they consumed warm filtered water; this is considered more effective than cold water at removing fatty dairy residues from the mouth [17]. The rationale for choosing a 5-min break was based on protein in saliva samples being considered to have plateaued within 3-min of WPB consumption (regardless of heating time) [33]. Saliva weight of all samples was measured by recording tube weight pre and post collection; all saliva samples were stored on ice pending analysis. The oral retention method development stages are outlined in Appendix B.

#### 2.3.3. Protein Analysis of Saliva Samples

In both studies, protein concentration (mg/mL) in saliva samples was analysed using Bradford Assay [51,52]. Samples were measured in triplicate with biological and analytical replicates using a 96 Well Plate Assay Protocol (Tecan Spark Control v2.1, Maneodorf, Switzerland). BSA was used as the protein standard, providing 6 decreasing dilutions mixed with purified water (SUEZ, Bristol, UK), ranging from 2 mg/mL to 0.125 mg/mL in triplicate, as well as a blank consisting of purified water on each individual 96 well plate. All saliva samples collected were analysed as a 1:2 dilution combining saliva and purified water with 5 µL pipetted into each well. Bradford Reagent (250 µL) was added to each well and each plate was placed on a shaker for 30 s and read within a 5 to 60-min period. All analysis was carried out immediately following a volunteer’s study visit. Each volunteers baseline saliva protein measurement (stimulated saliva protein concentration) was subtracted from their sample saliva measurements at each time point to calculate the concentration of protein remaining in saliva samples post beverage consumption.

#### 2.3.4. WPB Individual Perception and Liking

During the main study volunteers rated liking, effort to consume (easiness to drink and swallow), attribute perception and appropriateness of attribute level (Just-About-Right, JAR) of WPBs (WPCU and WPCH) (Figure 4) individually on an iPad (Apple, UK), either in isolated booths (younger adults) or at a table (older adults) using Compusense Cloud software (Compusense, ON, Canada). Samples, coded with three-digit random numbers, were provided in a monadic sequential balanced order, with sample sets randomly allocated to volunteers. Volunteers received 5 mL of WPB in opaque black plastic cups (25 mL) and all volunteers were trained by a short video as to how to use the generalised Labelled Magnitude Scale (gLMS), a scale ranging from no sensation (0) to strongest imaginable sensation of any kind (100) [53]. Volunteers had an enforced rest period of 45 s between samples and consumed warm filtered water before completing the same series of questions on the second sample.

### 2.4. Statistical Analysis

#### 2.4.1. Pilot Study

Statistical analysis was all carried out in SAS^®^ software (SAS Institute Inc., Version 9.4, North Carolina, NC, USA) using a linear mixed model considered robust enough for unbalanced data [54] and adjusted for multiplicity using Bonferroni. Saliva samples post beverage consumption were analysed using explanatory variables such as beverage type, time, gender and with volunteer code as a random effect and the dependent variable was protein concentration.

#### 2.4.2. Main Study

Tertiary analysis was used to categorise volunteers into low, medium and high groups, based on average salivary flow rates using XLSTAT (version 2019.2.2, Addinsoft, Boston, MA, USA); these groupings were also used for statistical analysis for unstimulated salivary flow rates. In order to test associations between age and categorical data (saliva flow rate grouping and medication) a chi-square test was carried out on contingency tables using XLSTAT. Statistical analysis was also carried out in SAS^®^ software using a linear mixed model and adjusted for multiplicity using Bonferroni. Analysis of saliva samples post WPB consumption used explanatory variables such as visit, age, beverage type, time, gender, saliva flow, medication and with volunteer code as a random effect and the dependent variable was protein concentration. Baseline saliva samples and salivary flow rates were analysed using explanatory variables of age, gender, visit and with volunteer code as a random effect and the dependent variables were protein concentration and saliva flow respectively. The data relating to volunteer WPB perception and liking was analysed using explanatory variables of age, gender, beverage type, saliva flow, medication and with volunteer code as a random effect and the dependent variables were gLMS data, liking and JAR scores. All attribute data which was collected on the gLMS log-scale was transformed to linear data (antilogged). Values are expressed as least square means (LSM) estimates, as these values best reflect the statistical model. Penalty analysis was carried out by XLSTAT with WPB JAR and liking scores, with 20% selected as the threshold for population size. Penalty analysis evaluated the influence of volunteer perception of appropriateness of attribute level rating (JAR) on volunteer liking by calculating the mean drop in liking rating (scale 1–9) compared with mean liking of volunteers that rated the attribute as JAR (JAR 3 on 1–5 scale), determined whether this drop in liking score is significant. Analysis of significant preferences between WPB samples was calculated using a Binomial expansion in V-Power [55]. In all analysis *p* < 0.05 was used as the value for significant difference.

## 3. Results

### 3.1. Pilot Study (Salivary Flow Rates and Saliva Samples Post Beverage Consumption) 

Salivary flow rates were 0.89 ± 0.33 mL/min and 2.56 ± 0.94 mL/min for unstimulated and stimulated saliva respectively. Beverage type (WPB or WPeB) had a significant effect (*p* < 0.0001), with the WPB leading to substantially and significantly more protein being collected in the saliva samples at all time points post beverage consumption (Figure 5). Time also had a significant effect (*p* = 0.0004), with saliva samples post WPB consumption showing a higher protein content at 15 s which decline over time (30, 60 and 120 s) whereas WPeB had a lower saliva protein content throughout which remained relatively constant.

### 3.2. Main Study

#### 3.2.1. Salivary Flow Rates

Older adults demonstrated significantly lower unstimulated saliva flow (*p* = 0.003) when compared with younger adults (LSM estimates ± SE: younger adults 0.90 ± 0.07 and older adults 0.62 ± 0.06 mL/min). However, age had no significant effect (*p* = 0.53) on stimulated saliva flow (younger adults 2.53 ± 0.19 and older adults 2.37 ± 0.18 mL/min). Volunteers were grouped by tertiary analysis into low, medium and high salivary flow rate, based on average salivary flow rates (Table 3). There was a significant association (*p* = 0.04) between age and saliva flow grouping for unstimulated saliva, however, stimulated saliva flow grouping was shown to be not significantly (*p* = 0.20) related to age. Age was significantly associated (*p* < 0.0001) with medication, indicating increasing use of medication with age (Appendix A) however, medication had no significant effect on saliva flow in older adults (unstimulated saliva flow (USF): *p* = 0.70 and stimulated saliva flow (SSF): *p* = 0.26) (data not shown). Gender had a significant effect (USF: *p* = 0.02 and SSF: *p* = 0.02) on saliva flow regardless of collection method; males having significantly higher salivary flow compared with females (USF: males 0.88 ± 0.07 and females 0.66 ± 0.05 mL/min and SSF: males 2.73 ± 0.20 and females 2.17 ± 0.15 mL/min).

#### 3.2.2. Saliva Samples Post WPB Consumption

Older adults had significantly higher protein concentration (*p* = 0.02) in their saliva samples compared with younger adults post WPB consumption, at all timepoints (Figure 6). Time had a significant effect (*p* < 0.0001) with most saliva samples post WPB consumption demonstrating a higher protein content at 15 s when compared with 30, 60 and 120 s. Although there was an overall significant difference (*p* = 0.05) between samples, with unheated WPB samples leading to a slightly higher protein concentration in saliva samples compared with a heated sample, this difference was not consistent at each time point and there were no significant differences between the samples at the timepoints (*p* = 0.14). There was significant variability between individual visits (*p* < 0.0001), but the overall trends remained the same (Appendix A). Although there was no overall significant effect of saliva flow (*p* = 0.06) on adhered protein concentration, the tendency was for the adhered protein content to decrease with increasing unstimulated saliva flow rate and this was significant at the 60 s collection time (*p* = 0.02) (Figure 7). There was no significant effect on protein concentration relating to gender and medication (Appendix A).

#### 3.2.3. WPB Individual Perception and Liking

The heated WPB was perceived as significantly (*p* < 0.05) thicker, less sweet and easy to swallow and resulted in more mouthdrying compared with the unheated WPB (Table 4). The increased thickness resulted in the beverage being significantly closer to “Just-About-Right” thickness as opposed to too thin (Table 5). There were no significant differences between the age groups, however, older adults did score attributes thickness and mouthdrying significantly lower than the younger adults for the heated WPB (Table 4). There was a significant effect (*p* = 0.03) of liking where older adults had significantly higher liking scores following the heated WPB compared with younger adults, however, there was no significant effect of liking on the unheated WPB. There was no significant effect of age on effort to consume and JAR attributes, though younger adults scored unheated WPB notable thicker than older adults (Table 4 and Table 5). There was no overall significant effect of saliva flow on WPB liking and perception, however, by categorising the volunteers into low, medium and high saliva flow groupings by tertiary analysis using unstimulated saliva flow, we found there were some interesting trends. In particular there was trends at a lower saliva flow for mouthdrying to be lower (low versus medium and high SF: *p* = 0.33 and *p* = 0.36 respectively), sweetness to be higher (low versus medium and high SF: *p* = 0.44 and *p* = 0.09 respectively) and thickness to be lower (low versus medium and high SF: *p* = 0.55 and *p* = 0.23 respectively) (Table 4). Volunteers not taking medication had significantly higher overall liking scores (*p* = 0.03) compared with medication users and males reported significantly higher easiness to swallow scores (*p* = 0.02) compared with females, however, no further significant effects on perception and liking were reported relating to medication or gender (Appendix A).

#### 3.2.4. Preference, Penalty Analysis and Qualitative Feedback

There was no significant preference (*p* = 0.46) between WPB samples, however preference was significantly influenced (*p* = 0.03) by age; younger adults preferred the unheated WPB, with older adults preferring the heated WPB (Appendix A). The volunteers’ perception of appropriateness of attribute level (Just-About-Right, JAR, ratings) can influence their overall liking, as shown in the penalty analysis (Table 5). There was a significant influence of thickness; where the older adults found the heated WPB to be too thin this led to a significant and substantial reduction in the liking rating. Volunteers’ comments were categorised into emerging themes, such as, flavour, texture, descriptive feedback, positive and negative comments and no comments provided (Appendix A). Overall, there was a general negative feedback provided for all WPB samples; volunteers provided comments relating to flavour and texture for both the unheated and heated WPB. In total 211 comments were provided with only 30 positive comments and the remaining 181 comments were all negative, some examples are summarised in Table 6.

## 4. Discussion

### 4.1. Mucoadhesion and WPB

The pilot study demonstrated that whey protein does adhere to the oral cavity (mucoadhesion) as the protein measured in the saliva samples following the consumption of a WPB was significantly and substantially higher than the protein content in saliva samples following consumption of a control whey permeate beverage (WPeB).These findings are supported by previous work which suggested proteins have adhesion and binding properties, for example: milk proteins can remain on oral surfaces [31], WPB can bind to oral epithelial cells [30], have mucoadhesive properties [32] and increased oral retention following a heated WPB compared with an unheated WPB [33]. However, these studies were carried out using animal models in vivo (measuring adhesion of proteins to porcine oral mucosa), with small subject sample sizes in human studies (5 volunteers) or without a non-protein source control. The pilot study demonstrated that our oral retention method is an effective and valid method to measure mucoadhesion in a WPB model.

### 4.2. Salivary Flow Rates

Older adults had, on average, a 27% lower unstimulated salivary flow rate when compared with younger adults. These findings are supported by Vandenberghe-Descamps et al. who found that healthy older adults had 38.5% lower resting salivary flow when compared with younger adults and a 38% lower stimulated saliva; theirs results were independent of dental and medication status [5]. However, our study did not find any difference in stimulated salivary flow rate between younger and older adults. It should however be noted that Vandenberghe-Descamps et al. measured unstimulated and stimulated saliva over a 10 and 5-min time period [5] compared with our study which used a 5 and 2-min time period which could have caused a greater difference between age groups. Age related changes to saliva flow are considered to relate to the submandibular and sublingual glands, which provide 70% of unstimulated saliva but less than 50% of stimulated saliva, providing a rationale for the greater reduction in unstimulated saliva compared with stimulated saliva in older adults [48].

Almost all of our study volunteers lacked experience in saliva collection, accordingly stress and behavioural factors could have contributed to poor adherence [56], for example, embarrassment about spitting, particularly in an unfamiliar setting. Our volunteers reported collecting stimulated saliva easier than unstimulated saliva therefore, stimulated saliva could be considered a more robust and representative measure compared with unstimulated saliva. As some volunteers struggled to produce unstimulated saliva, despite being considered healthy, future studies should consider familiarisation sessions before collecting such samples. Poor oral clearance is associated with reduced saliva function [7] and therefore could potentially explain the cause of food debris in unstimulated saliva samples, which was more prevalent in older adults in our study. However, there are age related changes associated with saliva (reduced volume and altered composition) resulting in saliva being potentially less watery and more concentrated [9]. A key challenge is understanding the causes of high variability in saliva flow associated within and between groups, however both lifestyle and the ageing process are thought to be potential causes of reduced saliva flow [5].

### 4.3. Saliva Samples Post WPB Consumption

There was an age-related increase in protein concentration in saliva samples post WPB consumption, which supported increased adhesion to the oral cavity from mucoadhesion [32,33]. A link between increasing protein concentration and reduced salivary flow rates has been previously suggested [6], and this was demonstrated by our volunteers where a low saliva flow correlated with increased protein concentration, therefore suggesting increased mucoadhesion of the whey protein, however, this needs further investigation. There is evidence of increased salivary albumin concentrations associated with frail older adults [57]. Salivary albumin has a role within the oral cavity as a serum ultrafiltrate and can potentially leak into saliva secretions [58] and is therefore a further possible contributor to increased protein concentrations found within this study. Therefore, although from the pilot study we can conclusively report that the protein content in the oral cavity post WPB consumption is due to adhesion of the protein from the beverage; we cannot rule out the possibility that the difference between younger and older adults could result from differences in salivary proteins rather than differences in mucoadhesion of the whey protein. The role of saliva composition and the changes to its physical properties were not measured in our study, however our study did find reduced salivary flow rates with age. A reduced saliva flow can be associated with decreased lubrication, protection, oral clearance, mucosal surfaces hydration and coating abilities within the oral cavity [6,7,8,9] and could lead to strengthened mucoadhesion by increased tissue exposure to whey proteins and therefore more adherence and interactions from proteins within the oral cavity [33]. In addition, there is evidence of saliva protein concentration being influenced by stress, inflammation, infection, hormonal changes and circadian variation [59]. These factors could potentially explain the differences in protein concentrations between visits, it should be noted however the overall trend was not affected.

There were minimal differences in protein adhesion related to whey protein heat treatment (unheated and heated WPB) which was unexpected. Previously, greater adhesion had been found to correlate with heated WPB samples compared with unheated WPB in a study using a small sample size of younger adults [33]. It is unlikely that the minimal differences found were due to cross-over effects and build-up of protein in the mouth, as the pilot study demonstrated substantial and significant differences between WPB and WPeB adhesion levels. This suggests that crossover effects between samples were minimal and indeed volunteers were provided with warm filtered water as a palate cleanser between samples, as well as having a 5-min rest between samples to minimise such effects. We therefore conclude that whey protein does adhere to the oral cavity and that any difference in adhesion due to heating treatment of the protein is minimal. Consideration is also required into how the different collection timepoints (15, 30, 60 and 120 s) influence saliva samples as a result of oral processing. For example, decreased muscle strength and swallowing difficulties are associated with ageing [60,61]. In the context of this work this could have influenced how quickly an individual could swallow a 10 mL sample (which could be particularly relevant at the 15 s timepoint) and affect their ability to hold a sample in the mouth during the 10 s swill time and the gathering of saliva in preparation for spitting, especially relevant at 120 s.

### 4.4. WPB Individual Perception and Liking

Mouthdrying was reported in this study following both unheated and heated WPB consumption. The heated WPB resulted in significantly increased perception of mouthdrying and thickness and significantly reduced sweetness, which led to a reduction in easiness to swallow. These differences were potentially caused by increased particle size on protein denaturation [18]. There was no difference in liking nor preference between the unheated and heated samples, which is explained by the overall low liking scores and potentially lack of familiarity amongst the volunteers with the taste and flavours associated with protein beverages.

### 4.5. Saliva and WPB Individual Perception and Liking

Individual saliva flow rates did significantly influence perception and liking of products. It was expected that a reduced saliva flow would result in increased perception of mouthdrying, however, the trend proved inconsistent, as unexpectedly, individuals with medium or high salivary flow rated mouthdrying higher compared with those with low salivary flow. This does support previous research which indicate an increased particle size detection threshold with increased saliva production in semi solid foods [62]. This may suggest a hydration mechanism associated with mucoadhesion, where the lubrication ability of saliva will strengthen adhesion properties with a resulting perception of mouthdrying [39]. It would therefore be assumed that within a liquid model, such as WPB, a low salivary flow will have reduced lubrication properties and therefore reduced adhesion properties, with lower resulting mouthdrying intensity. Although current results provide only a trend, it should be noted that mouthdrying in this study was measured at a single point in time. Therefore, future work should focus on investigating mouthdrying over time to gain a better understanding into the role of saliva flow on perception.

### 4.6. Age Related Changes in WPB Individual Perception and Liking

Age related trends were found within the age groups, however in most cases these trends lacked significance between the age groups, apart from the heated WPB, where older adults had significantly higher liking scores compared with younger adults. It could be therefore suggested that both age groups perceived the differences within a similar range. It was hypothesised that younger adults would be less sensitive to mouthdrying [17], however, younger adults reported greater intensity of mouthdrying compared with older adults. It could be suggested that the cause of the minimal difference between age groups may relate to how our study measured perception rather than lack of differences between age groups. For example, Withers et al. measured mouthdrying using a paired discrimination test [17] and our study measured mouthdrying using a gLMS scale, therefore potentially the differences in the results between the two studies could relate to the sensitivity of the tests used. Older adults tend to be less sensitive to taste and tactile sensations [63,64] which supports the conclusion in our study where older adults reported ‘too little flavour’ more frequently than younger adults and ‘too little thickness’ as the JAR attributes. Older adults reported a significant preference for the heated WPB which was the thicker beverage which contrasted with the younger adults. Perception of fluid viscosity has been found to decline with age [64] potentially explaining why our older adults preferred a thicker beverage and providing a design pointer for products for older adults. 

## 5. Conclusions

Protein does adhere to the oral cavity to a greater extent following the consumption of a WPB compared with consumption of a WPeB, indicating mucoadhesion of whey protein to be a true phenomenon. Saliva samples post WPB consumption, regardless of the extent of whey protein heat treatment, demonstrated increased adhesion to the oral cavity in older adults compared with younger adults. Such increased mucoadhesion with age may contribute to dislike of whey protein beverages, potentially due to a prolonged drying sensation, leading to poor consumption. Therefore, by understanding the potential mechanisms involved in whey protein derived drying, products could be reformulated to be more acceptable. It was expected that a reduced salivary flow would strengthen mucoadhesion; this trend was present but inconclusive, indicating the need for further research in this area. Heating of WPB resulted in significantly increased mouthdrying and thickness perception and significantly reduced sweetness, when compared with the unheated WPB. It would be necessary to carry out further investigation to determine conclusively whether perception and acceptance are influenced by age and individual differences in saliva flow, as the lack of clear age-related trends could have related to the sensory analysis being carried out at a single time point and not over repeated consumption. Further research is needed to fully establish whether mouthdrying changes with repeated consumption, whether it results from increased mucoadhesion, and to confirm the influence of age and saliva flow on mouthdrying and mucoadhesion. The overall aim of this work is to increase acceptance of protein fortified beverages for older adults at risk of malnutrition and sarcopenia; simple beverages, such as WPB, can alleviate this problem, however they need to be acceptable and palatable.

## Figures and Tables

**Figure 1 nutrients-12-02506-f001:**
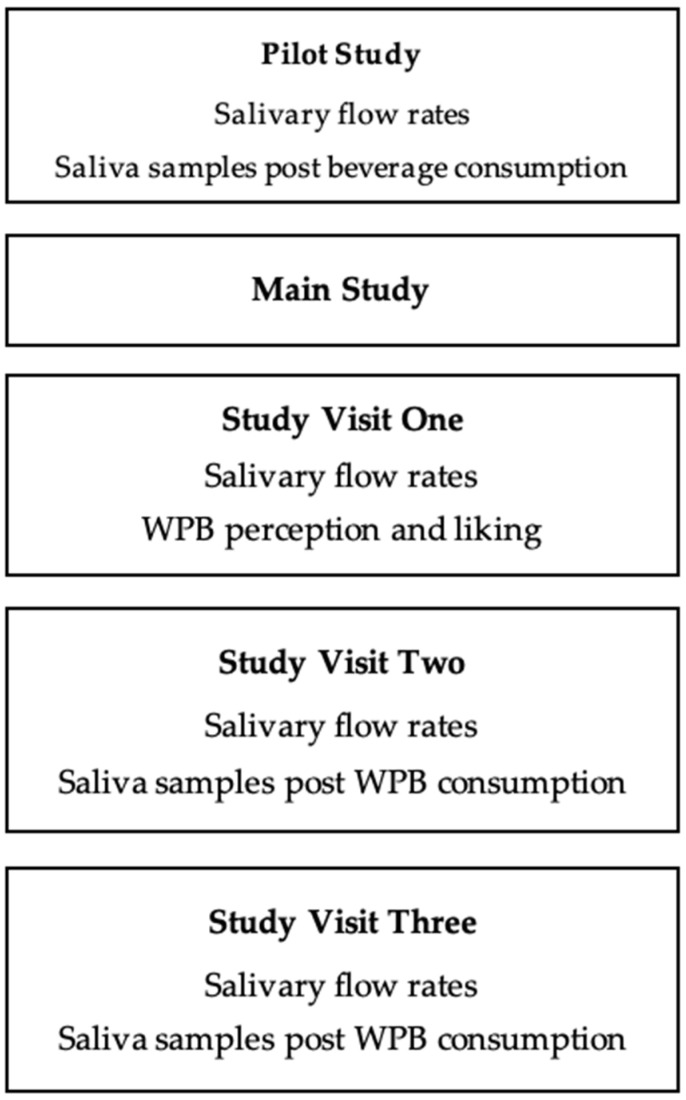
Overview of pilot and main studies (WPB: whey protein beverage).

**Figure 2 nutrients-12-02506-f002:**
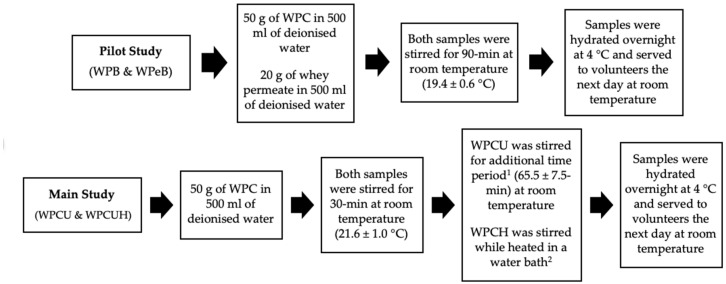
Overview of beverage preparation in both studies (WPB: whey protein beverage; WPeB: whey permeate beverage; WPC: whey protein concentrate; WPCU: unheated WPB; WPCH: heated WPB). ^1^ additional time period was based on the time it took to heat and cool WPCH; ^2^ the time to 70 °C was recorded (20.9 ± 4.7-min) and maintained at 70 °C for a further 20-min and cooled to room temperature.

**Figure 3 nutrients-12-02506-f003:**
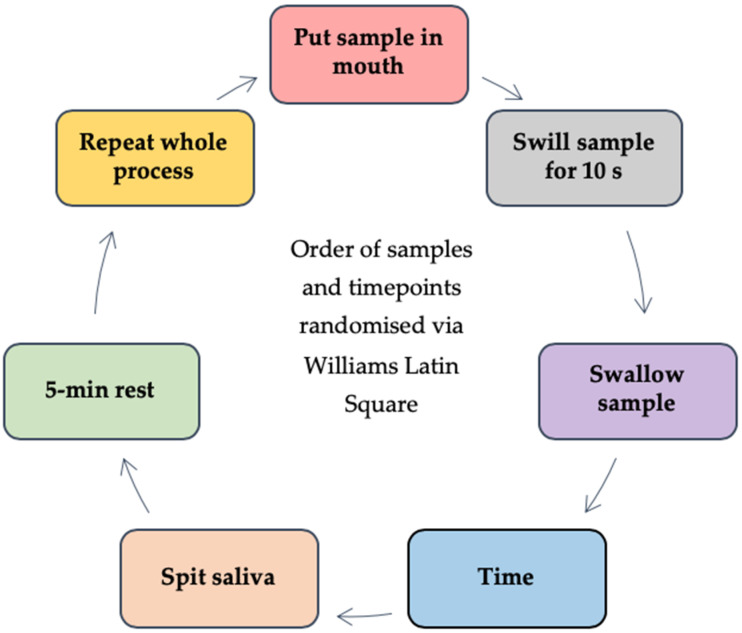
Brief overview of saliva samples post beverage consumption protocol. Volunteers were provided with verbal and written instructions as to the protocol and given the opportunity to ask questions. Volunteers were provided with one 10 mL sample and asked to swill the sample around in their mouth for 10 s before swallowing. After this a randomised countdown clock (time; either 15 s, 30 s, 60 s or 120 s) was started and once it reached zero, volunteers gave a saliva sample into wide lid collection tube (60 mL). A 5-min rest period followed, with the procedure being repeated for the seven remaining samples and timepoints.

**Figure 4 nutrients-12-02506-f004:**
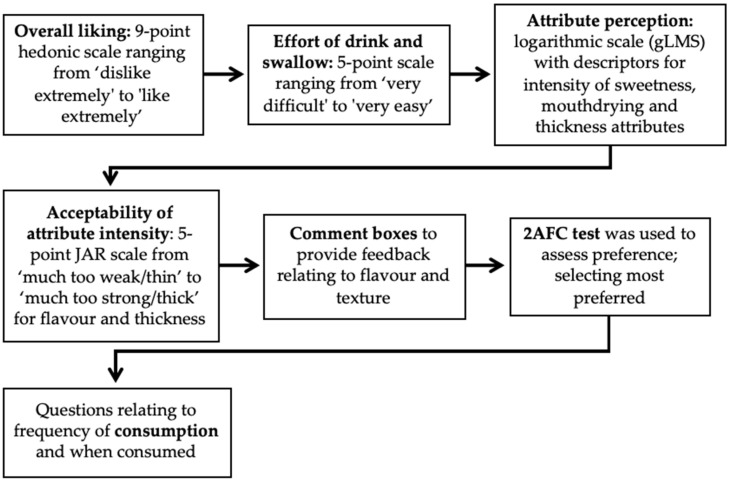
Overview of whey protein beverage (WPB) individual perception and liking (gLMS: generalised Labelled Magnitude Scale; JAR: Just-About-Right; 2AFC: two alternative forced choice).

**Figure 5 nutrients-12-02506-f005:**
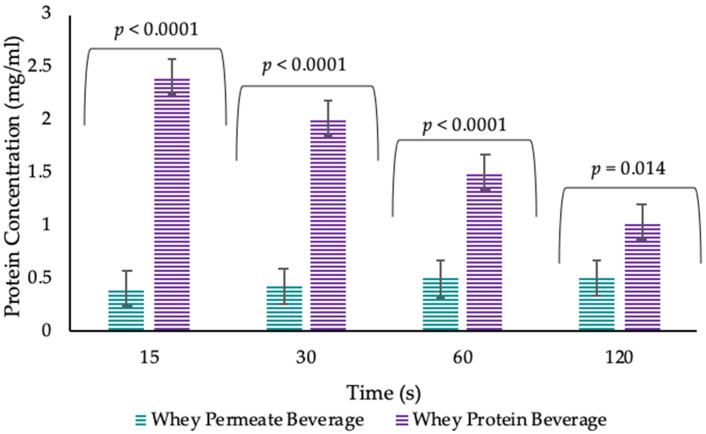
Protein concentration in saliva samples post beverage consumption by timepoints. Values are expressed as LSM estimates ± standard error from SAS output. Significant differences (*p* < 0.05) were reported between beverages at all timepoints with relevant *p* value above each timepoint.

**Figure 6 nutrients-12-02506-f006:**
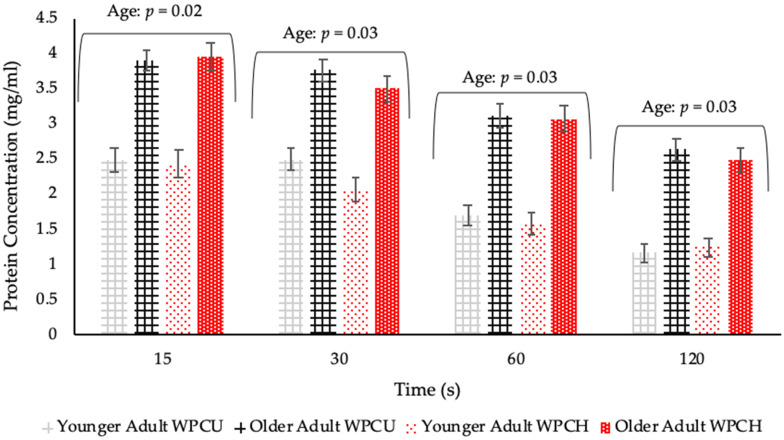
Protein concentration in saliva samples post whey protein beverage (WPB) consumption by age and timepoints (WPCU: unheated WPB; WPCH: heated WPB). Values are expressed as LSM estimates ± standard error from SAS output. Significant differences (*p* < 0.05) were reported between age groups at all timepoints with relevant *p* value above each timepoint. Data from visit 2 (n = 84; YA (Younger Adults) n = 42 and OA (Older Adults) n = 42) and visit 3 (n = 82; YA n = 40 (2 YA dropped out after visit 2) and OA n = 42) combined. Baseline saliva protein concentration values are outlined in Appendix A.

**Figure 7 nutrients-12-02506-f007:**
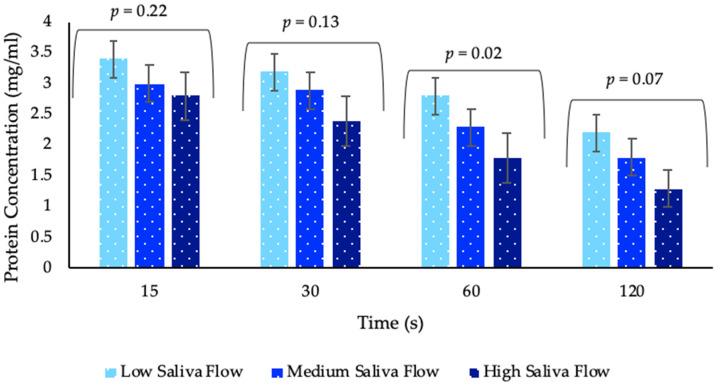
Protein concentration in saliva samples post whey protein beverage consumption by timepoints and saliva flow groupings. Values are expressed as LSM estimates ± standard error from SAS output. Significant differences (*p <* 0.05) were reported only at 60 s with relevant *p* value above each timepoint. Data from visit 2 (n = 84) and visit 3 (n = 82) combined. Individual saliva flow groupings are derived from unstimulated saliva flow only, through tertiary analysis.

**Table 1 nutrients-12-02506-t001:** Commonly proposed causes of whey protein beverage (WPB) derived mouthdrying.

Proposed Cause	Description
The pH of WPB	Low pH can cause precipitation of the protein, however, there is evidence of mouthdrying from WPB at both low and neutral pH [18,22,23,24,25,26]
Saliva and protein interactions	Perception of mouthdrying has links to saliva and protein interactions [22,23,25,27,28]
Reduced lubrication of saliva	Increased friction within the oral cavity from reduced lubrication [29]
Adhesion and binding properties	Whey proteins binding to oral epithelial cells, proteins remaining on surfaces, mucoadhesive properties and increased oral retention [30,31,32,33]
Heating time	Mouthdrying is considered to increase with product heating time, potentially due to protein denaturation [18,34]

**Table 2 nutrients-12-02506-t002:** Overview of whey protein beverage (WPB, 10% *w/v*) and whey permeate beverage (WPeB, 4% *w/v*).

	Whey Protein Beverage			Whey Permeate Beverage	
	Per 5 mL ^1^	Per 10 mL ^2^	Per 100 mL	Per 10 mL ^2^	Per 100 mL
**Nutritional Composition**					
Energy (kcal)	2.0	4.0	39.7	1.5	14.7
Fat (g)	0.04	0.07	0.7	0.0008	0.008
of which saturates (g)	0.01	0.03	0.3	-	-
Carbohydrate (g)	0.02	0.04	0.4	0.4	3.6
of which sugars (g)	0.02	0.04	0.4	-	-
Protein (g)	0.4	0.8	8.2	0.01	0.1
**Typical Mineral Profile**					
Calcium (mg)	-	5.5	55	2.2	21.6
Magnesium (mg)	-	0.5	5	0.4	4.4
Phosphorus (mg)	-	3.5	35	2.4	24.4
Potassium (mg)	-	5	50	5.7	57.2
Sodium (mg)	-	1.5	15	1.8	18.4
Chloride (mg)	-	0.5	5	1.8	18.4
**Chemical Properties**					
Protein %	-	0.8	8.2	0.01	0.1
Moisture %	-	0.05	0.5	0.004	0.04
Ash %	-	0.04	0.4	0.02	0.2
Lactose %	-	0.04	0.4	0.4	3.6
Fat %	-	0.07	0.7	0.008	0.008
pH	-	6.5	6.5	5.6	5.6

^1^ 5 mL sip size was used for WPB perception and liking in the main study only; ^2^ 10 mL sip size was used in the oral retention method in both studies.

**Table 3 nutrients-12-02506-t003:** Summary of volunteers salivary flow rates categories ^1^ (ml/min).

	Unstimulated Saliva Flow			Stimulated Saliva Flow		
	Low (0.04–0.53)	Medium (0.53–0.77)	High (0.77–2.18)	Low (0.23–1.63)	Medium (1.63–2.76)	High (2.77–6.13)
Total (n = 84)	28	27	29	25	30	29
Younger Adults (n = 42)	9	14	19	9	18	15
Older Adults (n = 42)	19	13	10	16	12	14

^1^ Categories are defined by tertiles with mL/min range for the category.

**Table 4 nutrients-12-02506-t004:** Volunteers liking, effort to consume and attribute perception mean ratings of whey protein beverages (WPB); overall and by age and unstimulated saliva flow rate.

	Overall (n = 84)		Age		Unstimulated Saliva Flow		
		Significance of Sample (*p* Value)	Younger Adults (n = 42)	Older Adults (n = 42)	Low Saliva Flow (n = 27)	Medium Saliva Flow (n = 28)	High Saliva Flow (n = 29)
**Overall Liking**							
WPCU	3.7 ± 0.3	0.10	3.6 ± 0.4 ^A^	3.7 ± 0.3	3.5 ± 0.4	3.8 ± 0.3 ^A^	3.6 ± 0.4
WPCH	3.3 ± 0.3		2.8 ± 0.4 ^aB^	3.9 ± 0.3 ^b^	2.9 ± 0.4	3.3 ± 0.3 ^B^	3.7 ± 0.4
**Easiness to Drink**							
WPCU	3.9 ± 0.1	0.11	4.0 ± 0.2	3.9 ± 0.2	4.2 ± 0.2 ^A^	3.8 ± 0.2	3.8 ± 0.2
WPCH	3.7 ± 0.1		3.6 ± 0.3	3.8 ± 0.2	3.7 ± 0.2 ^B^	3.8 ± 0.2	3.6 ± 0.2
**Easiness to Swallow**							
WPCU	4.2 ± 0.1	**0.0004**	4.4 ± 0.2 ^A^	4.1 ± 0.2	4.4 ± 0.2 ^A^	4.2 ± 0.1 ^A^	3.9 ± 0.2
WPCH	3.9 ± 0.1		3.7 ± 0.2 ^B^	4.0 ± 0.2	3.9 ± 0.2 ^B^	3.9 ± 0.1 ^B^	3.9 ± 0.2
**Mouthdrying**							
WPCU	16.9 ± 3.5	**<0.0001**	18.1 ± 5.2 ^A^	15.7 ± 3.8 ^A^	15.5 ± 4.8 ^A^	15.8 ± 4.6 ^A^	19.3 ± 5.0 ^A^
WPCH	28.0 ± 3.5		34.4 ± 5.2 ^B^	21.8 ± 3.8 ^B^	23.9 ± 4.8 ^B^	30.3 ± 4.6 ^B^	30.0 ± 5.0 ^B^
**Sweetness**							
WPCU	7.6 ± 1.1	**0.04**	7.1 ± 1.7	7.9 ± 1.2	8.7 ± 1.6	9.1 ± 1.5 ^A^	4.8 ± 1.6
WPCH	6.0 ± 1.1		6.4 ± 1.7	5.6 ± 1.2	8.2 ± 1.6	5.5 ± 1.5 ^B^	4.3 ± 1.6
**Thickness**							
WPCU	9.7 ± 2.0	**<0.0001**	11.5 ± 2.9 ^A^	7.9 ± 2.1 ^A^	9.5 ± 2.7 ^A^	9.9 ± 2.6 ^A^	9.7 ± 2.9 ^A^
WPCH	17.3 ± 2.0		19.5 ± 2.9 ^B^	15.2 ± 2.1 ^B^	13.3 ± 2.7 ^B^	18.2 ± 2.6 ^B^	20.6 ± 2.9 ^B^

Values are expressed as LSM estimates ± standard error from SAS output. Significant differences (*p* < 0.05) within a row (i.e., age YA vs. OA and saliva flow pairwise comparisons) are denoted by differing small letters and within a column (i.e., within an age group between samples or within saliva flow groupings between samples) are denoted by differing capital letters. WPCU (unheated WPB) and WPCH (heated WPB). Liking and effort to consume were measured on a 9- and 5-point scale respectively, attribute perception was measured on a gLMS logarithmic scale (antilogged values 0–100 scale presented). Individual saliva flow groupings are derived from unstimulated saliva flow only, through tertiary analysis.

**Table 5 nutrients-12-02506-t005:** Volunteers appropriateness of attribute level (Just-About-Right, JAR) mean ratings of whey protein beverages (WPB) and their influence on volunteer liking ratings; overall and by age (YA: younger adult and OA: older adult) and unstimulated saliva flow rate.

	Overall (n = 84)		Age		Unstimulated Saliva Flow			Penalty Analysis (Mean Liking Drop Where Attribute Deviates From Just-About-Right)			
		Significance of Sample (*p* Value)	Younger Adults (n = 42)	Older Adults (n = 42)	Low Saliva Flow (n = 27)	Medium Saliva Flow (n = 28)	High Saliva Flow (n = 29)	Too Little (YA)	Too Much (YA)	Too Little (OA)	Too Much (OA)
**JAR Flavour**											
WPCU	2.3 ± 0.1	0.29	2.4 ± 0.3	2.2 ± 0.2 ^A^	2.7 ± 0.2	2.0 ± 0.2	2.2 ± 0.2	1.04 *	2.18	1.21 *	1.07
WPCH	2.5 ± 0.1		2.5 ± 0.3	2.5 ± 0.1 ^B^	2.6 ± 0.2	2.1 ± 0.2	2.7 ± 0.2	0.59 *	0.71	0.80 *	0.63
**JAR Thickness**											
WPCU	2.2 ± 0.1	**<0.0001**	2.4 ± 0.2 ^A^	2.1 ± 0.1 ^A^	2.4 ± 0.2	2.2 ± 0.1 ^A^	2.0 ± 0.2 ^A^	1.06 *	0.80	0.56 *	−0.58
WPCH	2.7 ± 0.1		2.9 ± 0.2 ^B^	2.6 ± 0.1 ^B^	2.6 ± 0.2	2.6 ± 0.1 ^B^	2.9 ± 0.2 ^B^	0.73 *	0.29	1.17 ^a^	1.68 ^b^

Values are expressed as LSM estimates ± standard error from SAS output. Significant differences within a column (i.e., within an age group between samples or within saliva flow grouping between samples) are denoted by differing capital letters. Significant differences (*p <* 0.05) within a row (i.e., between penalty analysis groups within a sample for older adults) are denoted by differing small letters. WPCU (unheated WPB) and WPCH (heated WPB). Within Penalty analysis * represents where the size of the group is lower than 20% of the population. Individual saliva flow groupings are derived from unstimulated saliva flow only, through tertiary analysis.

**Table 6 nutrients-12-02506-t006:** Examples of volunteers whey protein beverage (WPB) comments (WPCU; unheated WPB and WPCH; heated WPB).

Sample	Comments and Volunteers Details
WPCU	*Tasteless and mouthdrying (v3, female, younger adult, aged 22). Bland flavour, unappealing colour, unsatisfying dry finish and aftertaste (v49, male, older adult, aged 65).*
WPCH	*It felt strange. It was thick and made my mouth feel dry afterwards. Almost as if all the moisture in my mouth had been sucked from it (v9, male, younger adult, aged 19). My mouth and teeth feel yucky. Like when you eat rhubarb, I would like to go and clean my teeth (v79, female, older adult, aged 75).*

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
