# Peer review of "An Investigation of the Influence of Age and Saliva Flow on the Oral Retention of Whey Protein and Its Potential Effect on the Perception and Acceptance of Whey Protein Beverages"

_nutrients, 2020, doi:10.3390/nu12092506_

Round 1
Reviewer 1 Report
The paper discusses the negative sensory attributes, mainly mouthdrying due to presence of protein (Mainly whey protein derived dryness), of protein fortifies drinks that used by older population to prevent sarcopenia. The influence of reduced salivation is discussed. It is a well written paper and some minor comments are given below.
Lines 92-103 – It will be better for the readers if the authors state the objective clearly first for each of the 2 points and then indicate what was done to achieve the goal.
Line 122 – Is ‘favourable opinion for conduct’ equivalent to approval of the studies?
Line 143, 146 – I think ‘ly’ in firstly and secondly are considered passé nowadays. First and second convey the same meaning
Line 248 – Just-About-Right, check throughout the manuscript
Line 251-252 – what is randomly allocated design? Clarify.
Line 256 – point out the reason for giving warm water.
line 283 – maybe rephrase the phrase – antilogged data. It would be useful if you mentioned a line about transformation of the data to linear scale.
Table 4 – in my opinion – superscripts should not be provided when the comparisons are not significant – it distracts the reader. That way it is easier to see the differences. Same view for all the tables.
Table 5 – the explanation of penalty analysis should be in methods section. Doesn’t have to repeated in the heading. Any explanation for the vacant cells?
Line 394 – is it preference? Or liking? Can they be used interchangeably?
Line 423 – that’s on the average right? The 27%?
Line 427 – was this 2.7% a significant amount?
Line 436 – did your panelists mention this in a debriefing session or you are just speculating?
Line 454 – when you say ‘Albumin’ – its not ‘Lactalbumin’ from the whey proteins in this case - right? This can be confusing to the reader.
Line 487 – any possibility that there was some astringency associated with these whey protein products that led to the mouthdrying? I see some references that mention astringency.
Line 531-532 – check sentence construction.
Reviewer 2 Report
please add a (although premature) speculation in your conclusion section in which way these results could contribute to a better acceptance of protein fortified beverages in the future.
